# Factors influencing self-management in relation to type 2 diabetes in Africa: A qualitative systematic review

**Joseph Ngmenesegre Suglo** [1]* , **Catrin Evans** [2]

**1** Department of Nursing, Presbyterian University College, Abetifi-Kwahu, Ghana, **2** School of Health Sciences, The University of Nottingham, Nottingham, United Kingdom

☉ These authors contributed equally to this work.

* suglojoseph@gmail.com

## Abstract

### Aim

Effective control of type 2 diabetes is predicated upon the ability of a person with diabetes to adhere to self-management activities. In order to develop and implement services that are locally relevant and culturally acceptable, it is critical to understand people's experiences of living with the disease. We synthesized qualitative research evidence describing the views and experiences of persons with type 2 diabetes in Africa regarding diabetes self-management.

### Methods

Five data bases (MEDLINE, EMBASE, PsychINFO, SCOPUS and CINAHL) were searched for qualitative studies published between the year 2000 and December 2019. After study selection, the included papers were critically appraised using an established tool. The data were extracted, and findings were coded and analysed to identify descriptive and analytical themes using a thematic synthesis approach. This review was registered in the international prospective register of systematic reviews (PROSPERO) with registration number CRD42018102255.

### Results

Sixteen studies were included in this review, representing a total of 426 participants across seven countries. Synthesis of findings produced six analytical themes. The diagnosis of diabetes triggered a range of emotions and revealed culturally specific understandings of the condition that negatively affected self-management practices. People with diabetes seeking health care at hospitals encountered several challenges including long waiting times and costly diabetes treatment. Family support and a state of acceptance of the condition were identified as facilitators to diabetes self-management.

**Data Availability Statement:** All relevant data are within the manuscript and its Supporting Information files.

**Funding:** The authors received no specific funding for this work.

**Competing interests:** The authors have declared that no competing interests exist.

## Conclusion

Effective self-management of type 2 diabetes is a challenge for most persons with diabetes in Africa. There is an urgent need for culturally appropriate education strategies and restructuring of the health system to facilitate self-management of diabetes.

## Introduction

Type 2 diabetes (hereafter referred to as 'diabetes') together with all other forms of diabetes results in approximately 1.5 million deaths globally [1]. It is a chronic metabolic condition of public health concern and one of the top four non-communicable diseases (NCDs) on the agenda of the United Nations (UN). Thus, Member States as part of the Sustainable Development Goals (SDGs) have set a target of reducing premature deaths due to NCDs by one third by 2030 [2]. This target calls for strategic policy intervention and proactive service delivery.

While 108 million adults were living with diabetes in 1980, this figure was estimated to have grown to 422 million by 2014 [1]. A further recent estimate by the International Diabetes Federation (IDF) indicates that approximately 425 million people (8.8%) between the ages of 20–79 years have diabetes, the majority from lower- and middle-income countries [3]. Africa has recorded an increasingly high prevalence of the disease. For instance, according to IDF [4] in 2015, 14 million people had diabetes in Africa and this figure is projected to rise to 34 million by 2040. This suggests that diabetes has reached epidemic levels in all parts of Africa. For example, in a meta-analysis, Hilawe et al [5] estimated diabetes prevalence to be 5.7% in Sub-Saharan Africa. Another population-based study of diabetes prevalence in African countries from 1980–2014 indicated that, diabetes in women and men had increased from 4.1% to 8.9% and 3.4% to 8.5% respectively [6]. According to IDF [3] in 2017, diabetes accounted for 298,160 deaths (6% of all mortality) in Africa. These alarming figures put pressure on the health systems of developing African nations who still experience the highest global prevalence of NCDs, including HIV [7], Tuberculosis and Malaria [8]. The direct and indirect cost of diabetes in the WHO's Africa region in 2000 was estimated to be US$67.03 billion, or US$8836 per person with diabetes per year [9], and some governments were spending up to 8% of their health expenditure on diabetes [10]. There is, therefore, a need for innovative management strategies to mitigate the growing impact of diabetes in the region.

Key elements in the management and prevention of diabetes and its complications involve life style modifications in areas such as exercise, diet, medication adherence, blood glucose monitoring and foot care, collectively referred to as 'self-management activities' [11–13]. Research in high income settings shows that following a structured diabetes self-management (DSM) approach leads to better clinical and quality of life outcomes [14]. Reports from a systematic review and meta-analysis in China indicates that, performance of self-management activities is influenced by diabetes knowledge, social support, household income levels and health beliefs [15]. Also, systematic review evidence from the UK demonstrates that psychological factors such as anxiety, depression, fear of hypoglycaemia and diabetes distress negatively affect DSM while social support, motivation and diabetes self-efficacy enhance an individual's DSM performance [16]. However, it is unclear whether this evidence around DSM experiences is transferable to the African context since existing research relevant to this region has not yet been synthesised. Reviewing the research on the perspectives of persons with diabetes will inform the development and implementation of services that are locally relevant and culturally acceptable.

## Aim

In order to understand the key factors that influence DSM, and thus to develop appropriate self-management models for sub-Saharan African countries, it is essential to understand people's views and experiences of managing their condition. Views and experiences are best explored through qualitative research [17, 18]. Hence, the aim of this review was to synthesise qualitative evidence that investigated the views and experiences of persons with type 2 diabetes in Africa regarding diabetes self-management (DSM). Specific objectives of the review were:

- To explore and describe the self-management experiences of persons living with type 2 diabetes in Africa.

- To describe self-management behaviours (exercise, medication adherence, diet, blood glucose monitoring and foot care) of people with diabetes.

- To identify factors that act as barriers and facilitators of self-management of type 2 diabetes, from the perspective of those living with the condition.

## Materials and methods

This qualitative systematic review is reported following the ENTREQ guidelines [19]. The review protocol (See S1 File) was prospectively registered in Prospero with Registration number CRD42018102255 [20].

### Search strategy

A preliminary search of Cochrane and JBI databases was done but we did not find any preexisting relevant systematic reviews on this topic. A comprehensive search strategy (See S2 File) was then constructed using key words and MesH headings in five databases: MEDLINE, CINAHL, EMBASE, PsychINFO and Scopus. The databases were searched from January 2000 to December 31st, 2019. In addition, hand-searches of the reference lists of all the included studies were undertaken.

### Inclusion and exclusion criteria

Table 1 gives details of the inclusion and exclusion criteria applied to the review. The WHO definition for 'Africa Region' (See S3 File) was utilised in study selection.

### Quality assessment

Study quality was independently assessed by two reviewers using the Joanna Briggs Institute Qualitative Assessment and Review Instrument (JBI-QARI) [21]. The appraisal enabled an indepth evaluation of the relative strengths and weaknesses of the included papers and how these may have influenced the synthesis [22–24]. See S4 File for quality appraisal of included studies.

### Data extraction

The JBI-QARI tool for data extraction was used to extract key study characteristics (methodology, methods, settings, geographical context, participants, phenomenon of interest, data analysis method and researcher's conclusions) and findings [25] (see S5 File for the data extraction template). The findings and discussion sections of the papers were extracted and coded to develop the synthesis. Data extraction was primarily undertaken by one reviewer (JNS),

**Table 1. Inclusion and exclusion criteria.**

| Variable | Inclusion | Exclusion |
|---|---|---|
| Population (Participants) | • Type 2 diabetes patients<br>• Adults above 18 years of age | • Type 1 diabetes patients<br>• Gestational diabetes<br>• Participants below age 18 years<br>• Non-diabetic patients |
| Phenomenon of interest | • Views and experiences of type 2 diabetes patients (beliefs, perceptions, attitudes, understanding, behaviours) regarding self-management of diabetes (exercise, diet, blood glucose monitoring, medication adherence and foot care) | • Research that investigated other aspects of type 2 diabetes patients other than their views and experiences regarding self-management |
| Context | • 'WHO Africa Region'<br>• Study setting includes homes or community settings and hospitals | • Studies conducted outside the 'WHO Africa Region' |
| Study Design | • Qualitative study of any design and the qualitative findings of mixed methods studies | • Other (non-qualitative) study designs |
| Language | • English | • Studies in other languages |

however the second author (CE) reviewed the data extracted for each paper and any discrepancies in interpretation were discussed.

## Synthesis

Synthesis of findings was done in three stages as outlined by Thomas and Harden [26]: (i) systematic coding of the results of individual studies; (ii) grouping of codes together based on similarity in meaning or shared characteristics to form descriptive themes, and, (iii) interpretation of higher order analytical themes. In order to maximise rigor within the process, we ensured that all themes in the synthesis were well supported by in-depth quotes from the studies. In addition, we sought to identify any disconfirming cases which were used to challenge and deepen our analyses [27].

## Results

### Search results

The database searches resulted in a total of 4,633 studies after removal of duplicates. Screening of titles, abstracts and reference lists of relevant studies resulted in 22 papers being retrieved for full text review. Six of these studies were later excluded because they did not meet the inclusion criteria based on the study design and population used. A total of 16 studies met the inclusion criteria. The search results are presented in PRISMA flow diagram Fig 1.

### Characteristics of included studies

The review included sixteen primary studies from seven African countries. Five of the studies were from South Africa [28–32]; four from Ghana [33–36]; two from Uganda [37, 38] and one each from Senegal [39], Ethiopia [40], Zimbabwe [41], Kenya [42] and Cameroon [43]. The total number of participants was 426. Participants were recruited from hospitals [30–32, 37, 40], diabetes clinics [34, 35, 38, 41], primary health care centres [28, 43], the community [29, 32, 33, 42] and medical clinics [39]. All studies were published within the last fifteen years (2003–2017) and the majority of the participants were females (n = 271, 64%). Two studies focused on culture in relation to self-management [39, 42], two studies explored management challenges from the perspective of persons with diabetes [28, 31] and five studies focused on

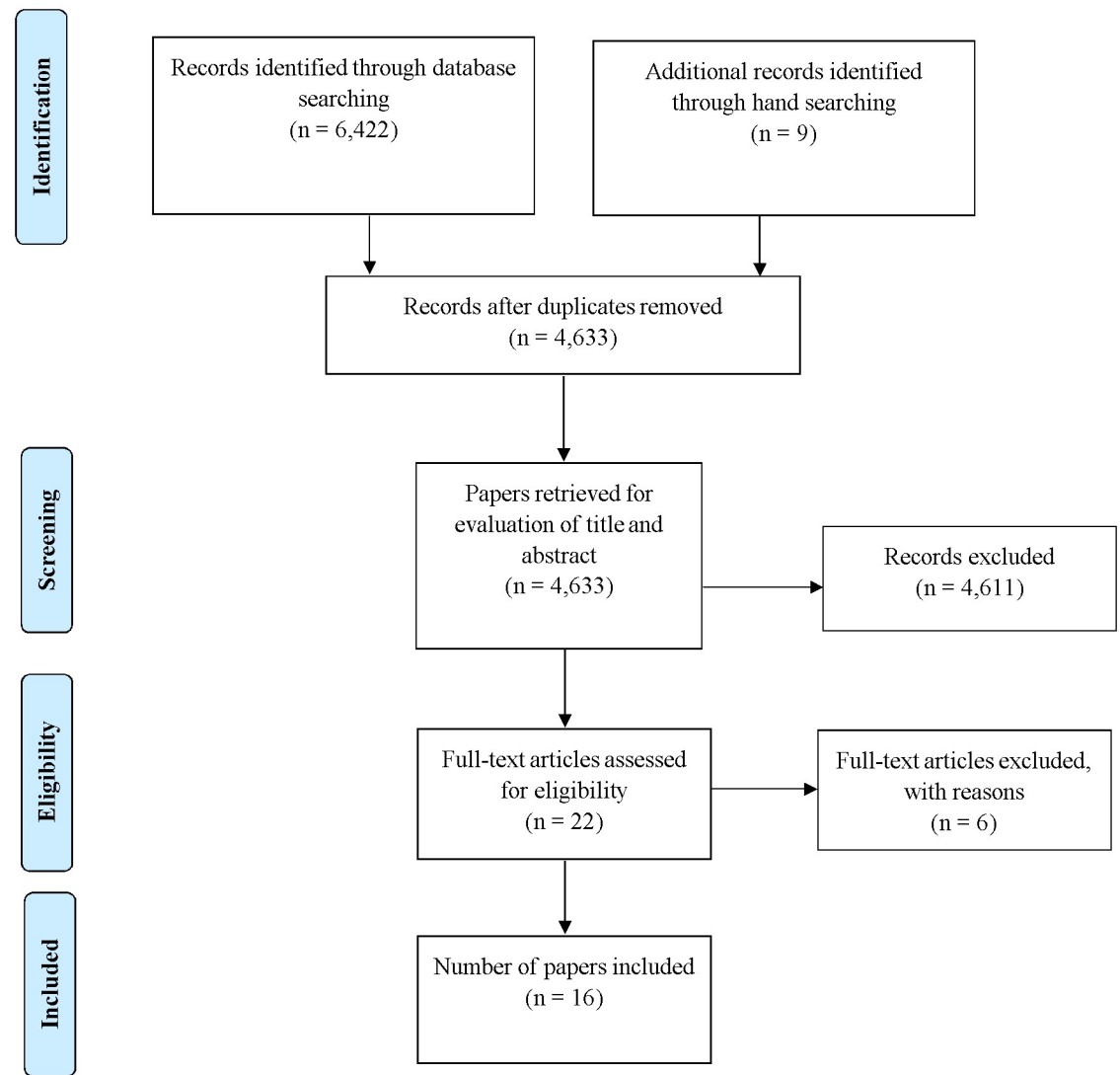

**Fig 1. PRISMA flow diagram (indicating the searching and selection of studies).**

participants general experiences of diabetes care [29, 30, 33–35]. Three studies focused on the beliefs about health and illness by persons with diabetes [37, 38, 41] and two studies looked at self-care practices of people with diabetes [36, 40]. The other two studies explored explanatory models to diabetes and compliance with biomedical treatments [33, 43]. Table 2 presents the characteristics of studies included.

## Methodological quality

The majority of the included studies were of sound methodological quality [21]. A particular strength of all the studies was a clear congruity between the research methodology and methods (questions two and three of the JBI tool). A weakness of several studies [29, 30, 39] was that the authors did not identify their own cultural and theoretical stand points (questions six and seven on the QARI tool). This lack of reflexivity limited our ability to assess the researchers' possible influence on the findings presented. This is especially important with cross-

**Table 2. Characteristics of included studies.**

| Study Reference and Country | Study Title | Setting | Participants | Data collection | Data analysis |
|---|---|---|---|---|---|
| Abdulrehman et al. [42] Kenya | Exploring the cultural influences of self-management of diabetes in coastal Kenya: An Ethnography | Community | Diabetic patients above 18 years. N = 30 (16 females) | Ethnographic interviewing, participant observation, and field note taking | Manual content analysis |
| BeLue et al [39] Senegal | A cultural lens to understanding the experiences with type 2 diabetes self-management among | Medical Clinic | N = 54 (35 females) | Semi- structured interview discussions guided by the PEN-3 cultural model | Content and PEN-3 analyses |
| Steyi and Phillips [28] South Africa | Management of type 2 diabetes mellitus: adherence challenge in environments of low socio-economic status | Primary health care | N = 26 (15 females) From six primary health care centers | Focus group discussions | Content analysis and identification of themes |
| O'Brien et al. [29] South Africa | Self-management experiences of persons living with diabetes mellitus type 2 | Community | N = 19 (13 females) | Semi-structured interviews | Tesch's method of thematic analysis |
| Mendehall and Norris [30] South Africa | Diabetes care among urban women in Soweto, south Africa: a qualitative study | Hospital | N = 27 (all females) | Face-to-face interviews | Content analysis |
| Adeniyi et al. [31] South Africa | Diabetic patients' perspective on the challenges of glycaemic control | Hospital | N = 17 (11 females) | Semi structured interviews using open ended questions | Thematic content analysis |
| Matwa et al [32] South Africa | Experiences and guidelines for footcare practices of patients with diabetes mellitus | Community, Hospital | N = 15 (10 females) | In-depth interview participant observation and field notes taking | Content analysis |
| Tewahido and Berhane [40] Ethiopia | Self-care practices among diabetes patients in Addis Ababa: A qualitative study | Outpatient department of 2 public hospitals | N = 13 (7 females) | Semi-structured interview | Thematic analysis |
| Hjelm and Beebwa [37] Uganda | The influence of beliefs about health and illness on foot care in Ugandan persons with diabetic foot ulcers | University Hospital | N = 14 (10 females) | Semi-structured individual interviews | unclear |
| Hjelm and Nambozi [38] Uganda | Beliefs about health and illness: a comparison between Ugandan men and women living with diabetes Mellitus | University hospital clinic | N = 25 (15 females) | Thematic interview guide with open ended questions | Qualitative content analysis |
| Hjelm and Mufunda [41] Zimbabwe | Zimbaween diabetic belief about health and illness: an interview study | Hospital diabetes clinic. | N = 21 (11 females) | Semi-structured interviews. | Qualitative content analysis |
| Awah et al. [43] Cameroon | Cure or control: complying with biomedical regime of diabetes in Cameroon | Primary health care facilities | Type II diabetes patients married or widowed N = 20 (9 females) | Focus group discussions, in-depth interviews, fieldwork conversations, and case studies | Content analysis which was inductive and continuous. |
| Doherty et al [36] Ghana | Type 2 diabetes in a rapidly urbanizing region of Ghana, West Africa: a qualitative study of dietary preferences, knowledge and practices | Tertiary hospital | 30 diabetes patients (20 female) | Focus group discussion and individual interviews | Themes were identified and coded using Nvivo10 software |
| de-GRAFT Aikins [35] Ghana | Living with Diabetes in Rural and Urban Ghana: A Critical Social Psychological Examination of Illness Action and Scope for Intervention | Medical clinics in Ghana | Diabetes Type I and II patients. N = 28 (14 female) | Semi-structured individual interviews were used | Atlas-ti qualitative data analysis package was used. |
| de-Graft Aikins [34] Ghana | Healer shopping in Africa: new evidence from rural-urban qualitative study of Ghanaian diabetes experiences | Rural and urban medical settings | Diabetes patients. N = 67. (41 Female) | Individual interviews, group interviews, and ethnographies. | Coding was done and Alas-ti qualitative analysis package used. |
| de-GRAFT Aikins et al. [33] Ghana | Explanatory models of diabetes in urban poor communities in Accra, Ghana | Community | Type II diabetes patients. N = 20 (18 females) | Individual interviewing | Thematic analysis guided by explanatory model of disease' concept. |

cultural research where language and cultural differences need to be considered explicitly as part of the research process. See S3 File for full details of the quality assessment of each included paper.

## Thematic synthesis of findings

A total of 170 initial codes were generated from all papers. These codes were grouped together based on similarities of meaning to form 13 descriptive themes. These themes were related to each other and explored to establish wider patterns, relationships and phenomena that influenced the experiences and behaviours of persons with diabetes. This led to the interpretation of six higher order analytical themes. Table 3 presents the descriptive and analytical themes, and their constituent studies. The themes were then further considered in order to infer potential barriers and facilitators of DSM as indicated in same Table 3. A narrative account of the analytical themes is presented below and supporting verbatim quotations to themes are provided in Table 3.

**Analytical theme one: Emotional reactions.** This theme describes the psychological and emotional experiences and expressions of persons with diabetes following the diagnosis. Findings from eight studies contributed to this analytical theme [29, 32, 34, 35, 37, 38, 41, 42]. It was formed out of two descriptive themes ('fear and denial of the diagnosis and implications of living with diabetes' and 'persons with diabetes accepted their condition gradually'). Following the diagnosis of diabetes, many participants expressed fear and denial of the diagnosis and others viewed it as a death penalty [29, 38]. Their anxiety and uncertainty were related to whether they would be able to cope with the consequences of diabetes such as taking medications every day, dietary restrictions, regular visits to the clinic and coping with the complications of the disease. Some participants who acknowledged their diagnosis of diabetes but were being treated with only tablet medications, perceived that their diabetes was not severe, and so were mostly reluctant in taking their medications [29, 32].

In some cases, participants who had previously lived with relatives with diabetes and had witnessed its complications, expressed anger and fear that they may also develop diabetes complications [29, 32, 41]. Nevertheless, some participants gradually accepted the diagnosis and took responsibility to manage it [29, 32]. Thus, the acceptance and self-management of diabetes only came with time for some participants. Nevertheless, though some participants accepted their diagnosis, they perceived diabetes as associated with chronic suffering and high medical expenses. In some cases, this led to disillusionment and inaction [34].

**Analytical theme two: Cultural beliefs on the causes and treatment of diabetes.** This theme relates to lay perceptions of the causes of diabetes and how this influenced the forms of treatment that was sought. Ten primary studies addressed this theme [31–35, 38, 39, 41–43]. This theme was derived from two descriptive themes as follows ('cultural beliefs and perceptions on causes of diabetes' and 'beliefs on using herbal and orthodox medicine to treat diabetes').

There were mixed responses with regards to what causes diabetes. Some participants were able to identify risk factors for diabetes such as hereditary or poor dietary habits [33, 35, 41, 42]. Caring for oneself as a person with diabetes included taking good food and medications prescribed by the doctor. Participants from studies in which recruitment had taken place in hospitals [32, 35, 38, 40, 41] appeared to have better knowledge on diabetes, suggesting that they may have received some form of education on diabetes.

However, a significant proportion of participants across the studies also held more traditional non-biomedical beliefs, believing that diabetes was variously caused by punishment from the gods, witchcraft, transmitted by mosquito bites or sexual intercourse [32, 34, 42, 43,

**Table 3. Themes, constitutive studies and inferred facilitators and barriers to DSM.**

| Analytical themes | Descriptive themes | Study reference | Illustrating Verbatim quotes | Facilitators to DSM | Barriers to DSM |
|---|---|---|---|---|---|
| **Emotional Reactions** | Fear and denial of the diagnosis and implications of living with diabetes | O'Brien et al [29], Hjelm and Nambozi [38], Hjelm and Mufunda [41], Hjelm and Beebwa [37], Abdulrehman et al [42], de-Graft Aikins [35] | *I thought I was going to die. . . I got scared, got worried, because I didn't know how I was going to look after myself. I begged doctor to tell me what I was suffering from, for I didn't believe it was DM. . . '.* *(Hjelm and Nambozi 2008:437 female 18).* <br><br> *"the diagnosis was six years and I think I'm still in denial, and I think the whole thing is probably you don't perceive it as being related to yourself"* (O'Brien et al. 2015:109) | • Persons with diabetes acceptance of their condition <br> • Diabetes perceived to be severe and serious <br> • Persons with diabetes accepting responsibility in caring for themselves <br> • Understanding diabetes as a manageable disease | • Denial of the diagnosis of diabetes <br> • Fear and anxiety <br> • Frustration with the diagnosis of diabetes <br> • Diabetes perceived as not severe <br> • Diabetes viewed as a death penalty and cannot be managed |
| | persons with diabetes accept their condition gradually | Matwa et al [32], O'Brien et al [29], de-Graft Aikins [34] | *As from then I accepted the fact that I had diabetes for life. And once you accept that, you become even more open to advice."* <br><br> *"you've got to acknowledge your illness. You've got to! If you don't acknowledge your illness, things won't go right, it will go wrong"* *(Matwa et al 2003:8)* | | |

*(Continued)*

**Table 3.** (Continued)

| Analytical themes | Descriptive themes | Study reference | Illustrating Verbatim quotes | Facilitators to DSM | Barriers to DSM |
|---|---|---|---|---|---|
| **Cultural beliefs on the causes and treatment of diabetes** | Cultural belief and perception on causes of diabetes | Abdulrehman et al [42], Matwa et al [32], Hjelm and Mufunda [41], Adeniyi et al [31], de-Graft Aikins [35], de-Graft Aikins [34], Awah et al [33], Awah et al [43] | *'yes, you can infect someone. Because I did not have diabetes, my husband is the one who had diabetes. Why did I then get diabetes?'* (Abdulrehman et al., 2016:5) <br><br> *"""The pastor took me through prayers and concluded that this diabetes is not mine but was bought and given to me".* [de-Graft Aikins 2015: R17] <br><br> *'So, if it was diabetes why didn't that doctor help it, and yet a black man did?' but let me warn you, never leave your nails lying around, the witches use the nails to make people develop ulcers'* (Matwa et al., 2003:16) <br><br> *'if it were not for my 'inyanga' (traditional healer), I would not be talking to you right now'* (Matwa et al. 2003:16) | • Belief that diabetes is caused by eating wrong foods <br> • Belief that diabetes is hereditary <br> • Belief that orthodox medicine can help manage diabetes <br> • Having a demonstrable knowledge on diabetes | • Belief that diabetes is caused by witchcraft <br> • Belief that diabetes is sexually transmitted <br> • Belief that real medicine is not free <br> • Misinterpretation that the symptoms exhibited is HIV or malaria and not diabetes <br> • Resorting to prayers in order to cure diabetes <br> • Depending on herbal medicine and faith healers <br> • Belief that diabetes can be cured completely |
| | Beliefs on using herbal and orthodox medicine to treat diabetes | Matwa et al [32], Belue et al [39], de-Graft Aikins [35], de-Graft Aikins [34], Awah et al [43] | *". . .If the insulin in the clinic was effective, I will not be turning to traditional medicine. . . . One alternate because one is desperate to obtain a cure for diabetes"(Awah et al 2008 page 5, 54-year old female.* <br><br> *". . .I mixed traditional plants and western medications to treat my diabetes. My blood sugar used to be as high as 5 grams; because of my medication, now it is only 2 grams."* (Belue et al 2013:337:65-year-old female) <br><br> *"I want to be healed so I will follow up whenever I hear of somebody who can help"* (de-Graft Aikins 2003). RF02 page 12. | | |

*(Continued)*

**Table 3.** (Continued)

| Analytical themes | Descriptive themes | Study reference | Illustrating Verbatim quotes | Facilitators to DSM | Barriers to DSM |
|---|---|---|---|---|---|
| Social obligations, relationships and support for persons with diabetes | Support from family and significant others | Belue et al [39], Stayl and Phillip [28], O'Brien et al [29], Hjelm and Mufunda [41], Mendenhall and Norris. [30] | "I get a lot of support from my family and from my friends. . . My son who was born in 1974 is a grown man and he helps me a lot financially. The rest of my family supports me mentally. My wife and my daughter-in-law cook my food." (Belue et al., 2013:334, 57-year-old male).<br><br>"To say the least, my boss is very strict and does not allow you to eat whilst you work. I work in a factory. Even though I have a letter from the hospital to say that I must eat regular small meals, he insists that I "clock in and out". Now I lose pay!" (Stayl and Phillip, 2014:4). | • Having other relatives who understand the disease<br>• Supportive family members<br>• Supportive friends | • Cooking food for a whole family<br>• Attending weddings and social gatherings<br>• Religious observances (fasting)<br>• Pressure from friends<br>• Living with extended family members<br>• Gender role of cooking (for women) |
| | Social obligations of persons with diabetes | Abdulrehman et al [42], Tewahido and Berhane [40], de-Graft Aikins [35] | "most likely when I attend social events, it is to please the hosts and not myself. . . . the reason is when people drink soda, I do not. . . . in Islamic religion to attend social events when you are invited is an obligation." (Abdulrehman et al., 2016:8).<br><br>"I can't go to a 'mahber' (social event) for instance and say 'I won't eat or drink'. I take what they give me with pleasure because it is not appropriate to refuse, as the saying goes, 'yeweledutin kalsamulet, yakerebutin kalbelulet', (a guest is disrespectful. . .if failed kissing the host's children or if refused eating food served by the host) therefore I go and I eat what they have prepared. . ." 56-year-old female(Tewahido and Berhane 2017:5) | | |
| | Sexual function and relationships | Hjelm and Nambozi [38], O'Brien et al [29], de-Graft Aikins [35] | ". . . I could have worked to buy drugs regularly. Now I cannot work" (de-Graft Aikins, 2003, RM04), Page 10.<br><br>'My private parts are weak. . . I no longer function sexually. . . (and) I am unable to meet the needs of financial problems' (Hjelm and Nambozi, 2008:438)<br><br>"I hate it when I go everywhere, and people force you." (to eat what they are eating) (O'brien et al. 2015: 8) | | |

(Continued)

**Table 3.** (Continued)

| Analytical themes | Descriptive themes | Study reference | Illustrating Verbatim quotes | Facilitators to DSM | Barriers to DSM |
|---|---|---|---|---|---|
| Self-management practices of persons with diabetes | Dietary management of diabetes | Belue et al [39], Hjelm and Nambozi [39], Matwa et al [32] Mendenhall and Norris [30], Tewahido and Berhane [40], Doherty et al. [2014], de-Graft Aikins [2003] | *'being African, I eat rice for lunch because it is part of my culture…even though we know it is affecting our health in a negative way'* (Belue et al 2013:335. 53-year-old female) <br><br> *"It is quite clear that whenever you go off the diet programme, then you're in trouble…* (de-Graft Aikins 2003, UM04 page 12) <br><br> *"Local foods are healthier than packaged because of the sugar content. If you want to live long, avoid packaged foods"* (Doherty 2003, 46-year-old urban male) | • People with diabetes awareness of the significance of diet in the management of diabetes<br>• Using advise from diabetes class<br>• Belief that healthy diet is important for health<br>• Compliance with recommended self-care practice<br>• Self-discipline to follow recommendations<br>• Adjusting drug dosages to take care of blood sugar<br>• Medications perceived as the most important and source of survival<br>• Readiness to take diabetes medications<br>• Exercise through routine daily activities<br>• Exercise through daily household chores<br>• Ability to use body signs to assess blood glucose levels<br>• Persons with diabetes used signs and symptoms to detect changes in health status (Blood sugar level)<br>• Ability to recognise signs of hyper and hypoglycaemia<br>• Knowledge on foot care<br>• Washing of feet | • Deliberate non-compliance with dietary recommendations<br>• Diabetic diet being repetitive and boring<br>• Diabetic diet being restrictive<br>• Cultural beliefs and attachment to some foods<br>• Costly nature of diabetic foods<br>• Multi-drug therapy for other co-morbid conditions<br>• Unable to travel with medications<br>• Old age and physical inability to exercise<br>• Physical disability of persons with diabetes<br>• Shoulder and knee problems of persons with diabetes<br>• No suitable ground and space in the community for exercise<br>• Expensive gyms<br>• Lack of motivation to exercise<br>• Foot care not a recognised self-care practice<br>• Poor knowledge on Foot care |
| | Physical activity/exercise | Tewahido and Berhane [40], Mendenhall and Norris [30], O'Brien et al [29], Matwa et al. [32], Abdulrehman et al. [42], Steyl and Phillips [28] | *"Even if I was committed to regular exercise, it is not convenient. There is no place to exercise in the city and the gyms are not affordable."* (Tewahido and Berhane, 2017:5. 44-year-old male) <br><br> *"I don't exercise much but I make up for it with household chores and a bit of gardening"* (Mendenhall and Norris, 2015:5) | | |
| | Managing diabetes with medications | Tewahido and Berhane [40], Abdulrehman et al [42], Hjelm and Nambozi. [38], Matwa et al. [32]; O'Brien et al [29], Adeniyi et al [31], Awah et al [43], de-Graft Aikins [34] | *"I mostly follow the doctor's orders. But when it (blood sugar) is unacceptably high, let's say above 250, then I slightly increase the dose."* (Tewahido and Berhane, 2017:6. 58-year-old male) <br><br> *'I have purchased medicines that were fake… it is known that some (pharmaceutical companies) make fake drugs so that they can make lots of quick cash.'* (Abdulrehman et al., 2016. Pg. 9 column 1 | | |
| | Monitoring of blood sugar level | Hjelm and Nambozi [38], Abdulrehman et al [42], Tewahido and Berhane [40] | *"I do not have the requirements to use, but I use the signs when it is too high and low I know from the signs… passing a lot of urine and drinking a lot. When I am sleepy, I know it is low".* (Hjelm and Nambozi, 2008:438. Female 2.). | | |
| | Foot care | Abdulrehman et al. [42], Tewahido and Berhane [40], Matwa et al. [32], Hjelm and Beebwa [37] | *"… I also like putting powder in-between my toes because I sweat a lot".* <br><br> *"my mother decided to soak it (the foot) in hot water. It became like cook meat in so much that some pieces of flesh fell off. The whole leg was rotten"* (Matwa et al 2003:18 | | |

*(Continued)*

**Table 3.** (Continued)

| Analytical themes | Descriptive themes | Study reference | Illustrating Verbatim quotes | Facilitators to DSM | Barriers to DSM |
|---|---|---|---|---|---|
| Economic impacts of diabetes | Financial challenges | Adeniyi et al [31], Steyl and Phillips [28], O'Brien et al [29], Matwa et al. [32], Abdulrehman et al. [42], Mendenhall and Norris [30], Belue et al. [39], Hjelm and Mufunda [41], Tewahido and Berhane [40], de-Graft Aikins [35], de-Graft Aikins [34], Awah et al. [43] | 'Healthy food is a lot of money. . . actually, all foods are expensive. Pap and bread are the cheapest.' (Steyl and Phillips, 2014:4 Female, 71 years)<br><br>'now you only have one visit allowed in a year, if you have to go again, it is another four or five hundred rand out of your pocket.' (O'Brien et al. 2015:111)<br><br>"When prescriptions are made at the hospital, I have to wait for my children to buy them for me. . . ." (Awah et al. 2008: 6)<br><br>"When I am due to see the doctor but I am unable because of financial problems, I think a lot, because I am afraid that my situation will worsen" (de-Graft Aikins 2003: 11) | • Government funding persons with diabetes to visit the clinic<br>• Affordability of diabetes medication | • Poor economic situation of persons with diabetes<br>• Unemployment<br>• Being dependents on others<br>• Costly diabetes treatment and food<br>• Unsupportive managers/ employers |
| | Employment problems | Matwa et al [32] | At the clinic they punctured the blisters. But since then I have been in and out of hospital, both feet are raw, raw, raw; I had to stop working." (Matwa et al., 2003:17). | | |
| Health care system | Health care professionals and the health system | Hjelm and Mufunda [41], Adeniyi et al [31], O'Brien et al [29], Hjelm and Nambozi [38], Hjelm and Beebwa [37], Matwa et al [32], Steyl and Phillips [28] | 'they (health workers) give information. . .teach me those things that I am supposed to and not supposed to do. . .give more advice on food. . .inform on how to give injections' (Hjelm and Mufunda, 2010. Pg.5 column 1).<br><br>'No advice from the health care workers, sometimes, they are too much hurrying' (Adeniyi et al., 2015. Participant 06; F, 54 years)<br><br>'get the facts right themselves first, because a lot of them (nurses) haven't got a clue.' (about DM type 2) (O'Brien et al 2015:9). | • Health professionals able to educate persons with diabetes<br>• Nurses being knowledgeable on diabetes<br>• Availability of leaflets on diabetes available | • Long waiting time and queues at clinics<br>• Health professionals in a rush/ hurrying during consultation<br>• Costly and non-availability of treatment at the clinic |

45]. Others felt it must be malaria or HIV and did not believe it was diabetes. Misconceptions about the causes of diabetes seemed greater in studies conducted in rural communities [32, 42, 43]. Lay beliefs about the causes of diabetes led to a range of local diabetes management strategies such as resort to herbal medicine, traditional healers and prayers. For example, many of the respondents in the studies from Kenya, Cameroon and Ghana [34, 42, 43] stated a belief that herbal preparations could completely cure their diabetes. Furthermore, a majority of participants from a Ghanaian study [34] engaged in "healer shopping", moving from one doctor to the other or one traditional healer to the other in search for a definitive cure to their diabetes. Thus, diabetes was not necessarily understood as chronic lifelong condition and this made some participants who were on orthodox medications, still resort to herbalists seeking a cure.

**Analytical theme three: Social obligations, relationships and support for persons with diabetes.** This analytical theme refers to the type of support persons with diabetes received from their families and other members of the community and illuminates the ways in which traditional family roles and social obligations impacted on diabetes care [28–30, 35, 38–42]. The descriptive themes constituting to this theme includes ('support from family and significant others', 'social obligations of people with diabetes' and 'sexual function and relationships').

The studies suggested that the family had both negative and positive influences on participants DSM activities [29, 39]. Family members' roles were particularly significant in the areas of medication and dietary management of diabetes. For instance, many participants reported having to rely on the financial support of family members to purchase medications [35, 39, 41, 42]. Nevertheless, to some participants, family members were also a hindrance to their effective DSM especially in the area of diet. In most parts of Africa, the extended family system is cherished, and many households practice communal eating. In such settings, many of the persons with diabetes reported that food would be prepared for the whole family (by someone else) and hence they tended to eat whatever was prepared for the family even though such foods might not be suitable for people with diabetes [28, 30, 39].

Kinship and religion, thus, social and religious obligations of people with diabetes also affected DSM activities. Some participants were often required to participate in customary or ritual activities. For example, attending events such as a wedding feast when invited is an obligation in Islamic marriage, and fasting during the month of Ramadan is obligatory [42]. In addition, some participants reported having reduced libido or reduced sexual function [43]. This was reported by both men and women who were concerned that their inability to function sexually may cause relationship problems. All participants from a study in Ghana noted that symptoms of diabetes had caused them some disruption to their body including sexual dysfunction, visual impairment and physical disability [35]. This indicates a disruption to the body self and could affect their ability to work and carry out social roles.

**Analytical theme four: Self-management practices of persons with diabetes.** This theme describes the experiences and practices of participants in relation to the various components of DSM (exercise, diet, medication, blood glucose monitoring and foot care). All of the included studies contributed to this analytical theme except one [41]. This theme is derived from five descriptive themes (dietary management of diabetes, physical activity/exercise, managing diabetes with medications, monitoring of blood sugar levels and foot care).

Participants in most studies acknowledged that eating the right foods was necessary for controlling diabetes [30, 32, 36, 38–40]. However, eating the recommended diet was noted to be difficult because of the cultural values placed on some foods, as well as the costly nature of diabetic diet. For example, some participants in Senegal indicated that, as Africans, they eat rice everyday as part of their culture, even though they knew it could affect their health negatively [39]. Participants in some studies reported a deliberate non-adherence to recommended

diet as it was regarded as being too repetitive and restrictive [32, 39]. This was worst among respondents who indicated they have limited food choices in their local markets. Most study participants in Ghana [36] were engaged in eating bread and biscuits as meal substitutes and for convenience, even though this went against medical advice. However, some participants tried to follow dietary advice from doctors but were occasionally confused as to what quantity and type of food they should eat due to conflicting information from health workers [36].

Another area of DSM was exercise. Some participants demonstrated knowledge on the importance of exercise and this motivated them to follow a recommended exercise plan [29, 32]. There were several participants who enjoyed and preferred some specific types of exercises, but due to their physical inabilities like knee and shoulder problems, old age and mental incapability, they could not exercise regularly [28, 29, 40]. In addition, persons with diabetes living in urban areas noted that there was no convenient space to exercise and that gyms in the city were not affordable [28, 40]. Some participants from rural settings though did not engage themselves in formal exercise programmes, they regarded their engagement in farm activities and household chores as enough exercise [30, 40, 42]. Thus, long distance walk to the farm, visiting friends and other routine daily activities were seen as forms of exercise by rural dwellers.

Using medications to manage diabetes was clearly identified by many participants as the most important aspect of self-management [32, 34, 35, 38, 40]. This motivated participants to adhere to prescribed medications, though some reported adjusting their own dosages to be able to control their glycaemic levels [29, 32, 40]. However, some persons with diabetes identified practical concerns such as an inability to carry medications (especially insulin) when travelling which might lead to missed dosages [31, 40]. Thus, some persons with diabetes were unable to travel with insulin due to lack of storage facilities and whereas others simply feared the injection pain.

A study from Uganda [38] indicated that monitoring of blood glucose level was described by many participants as necessary for their health. However, other studies reported that the practice was uncommon and irregular [40, 42]. Due to financial constraints, most participants did not have a personal glucometer to monitor their blood glucose at home and they checked their blood sugar levels only when they visited the clinic. Participants without personal glucometers could not keep a diary of their blood sugar readings and sometimes rely on their bodily signs to guess their blood sugar levels [40, 42].

Only three papers reported on foot care practices of participants [32, 40, 42]. Recommended foot care practices such as inspection, caring for sores, abrasion and cracks seemed to be uncommon among participants, and though many reported foot problems, they had not sought specialist medical care. In a study in Kenya [42] participants noted that they were unable to wear the appropriate closed shoes because of the heat of the tropical weather and they could sustain injuries to their feet while working on their farmlands. Washing of feet was the only known foot care practice identified in Uganda [37]. Another study in South Africa reported that all participants regularly applied emollients and moisturisers to their feet after washing [32]. This practice was however mixed with some inappropriate practices like soaking the feet in hot water, applying hot water bottles and powder in-between toes.

**Analytical theme five: Economic impacts of diabetes.** This analytical theme represents the financial consequences of managing and living with diabetes. Ten studies contributed to this theme [28–31, 35, 36, 39–42], and their findings were categorised into two descriptive themes ('financial challenges' and 'employment problems').

Financial challenges in the management of diabetes was related mostly to the need to purchase medications and the right diabetic diet [28–31, 39–42]. Studies reported that participants tended to eat whatever they could afford and sometimes ran out of medication for days. Also, a study in South Africa noted that persons with diabetes were funded to receive only one

hospital visit per year, and the cost of any additional visits to the hospital diabetes clinic must be borne by the individual [29]. This made some people with diabetes to stay at home even when unwell. Diabetes and its complications were described by some participants as having a significant impact on their income sources. For example, due to diabetic foot problems, some participants needed to make frequent visits to the hospital at the expense of their job, and some had to stop work and depend on family and friends [32].

**Analytical theme six: Health care system.**   This theme explains how health workers' attitudes and the nature of service delivery affected self-management behaviours of persons with diabetes. Seven studies contributed to this theme [28, 29, 31, 32, 37, 38, 41]. Participants expressed mixed feelings about the health care system and how care professionals' attitudes affected the management of their diabetes. Participants regarded health care professionals as essential in the management of diabetes because they educated them on diet, medication and injection techniques [37, 38, 41]. However, participants also recounted many challenges they faced in seeking the services of health care workers. These included: doctors arriving late at the consulting rooms, nurses and doctors in a hurry and not giving participants enough time and explanations, long queues and waiting time at clinics which demotivated some persons from attending and non-availability of drugs at clinics [28, 31, 41]. Thus, some persons with diabetes never visited the diabetic clinic again after their initial visit because they alleged drugs were not available at the clinic and that healthcare workers were too busy to listen to all their concerns.

## Discussion

This thematic synthesis of the views and experiences of persons with diabetes has identified findings that resonate with conclusions reported in previous reviews that include studies from other geographical contexts, but has also identified findings that appear particularly salient to the African context. These features will be highlighted within the discussion below. Key findings of this review are discussed in two domains: (i) factors affecting DSM relating to the individual and community, and, (ii) factors affecting DSM relating to the health care system.

### Factors affecting diabetes self-management relating to the individual and/or community

In line with findings from other reviews on DSM [46, 47], our review showed that persons with diabetes experienced a range of emotional reactions ranging from denial to frustration. Thus, following the diagnosis of diabetes, some participants experienced a feeling of loss of their health and went through a "cycle of grief" [44]. The conflicting emotions of grief, shock, denial and frustration experienced following the diagnosis of chronic diseases are normal and people should be supported to make the necessary adjustment [45]. Persons with diabetes may be overwhelmed with the life style changes required in DSM and this can trigger negative emotions. Emotions of fear, anger and hopelessness have been identified among persons with diabetes all over the world [46, 47]. Such feelings, if not well managed, can lead to psychological disorders such as depression, anxiety and stress which negatively impact on DSM behaviours [29, 48]. Conversely, it has been suggested that for persons with diabetes, achieving more positive emotional states (such as contentment) results in better health behaviours and improved adherence to treatment regimens [44]. This indicates the significance of psychological and emotional wellbeing for diabetes self-care. It is imperative therefore that diabetes care interventions seek to promote positive psychological health [48]. The evidence base on interventions to promote psychological resilience and wellbeing in the context of chronic illness in Africa is limited, however it is likely that family and community focused initiatives will be important [48]. Health professionals caring for individuals with diabetes should, therefore,

engage in effective dialogue to provide psychological and emotional support to patients at different stages of their disease [48].

The review also showed that persons with diabetes used both herbal and orthodox medicines to treat their diabetes, which, in some cases, negatively affected DSM. Resorting to herbal treatment was mostly due to the misconception that diabetes can be cured completely and that it is caused by punishment from the gods or witchcraft. These ancestral and superstitious beliefs about causation of chronic diseases in Africa makes persons with diabetes reluctant to follow orthodox treatment regimens. Similar findings have also been reported in other developing nations. For instance, a systematic review in South Asia identified similar limited knowledge and misconceptions about diabetes as barriers to DSM [49]. However, irrespective of people's knowledge, individuals also use herbal medicines because it is a cultural norm, because it is cheap and because it is perceived to be effective [50]. Therefore, it is important that clinicians recognise the cultural and economic basis for why persons living with diabetes in Africa may still use herbal preparations even when they have been prescribed biomedical treatment. More research is needed to explore the potential for diabetes care algorithms to incorporate the use of clinically effective herbal preparations in order to give patients greater choice over potential treatment options. It is for this reason that the WHO regional office for Africa is actively encouraging the integration and use of herbal medicines in health systems in the African Region [51]. Indeed, through the advocacy and efforts of WHO, 36 countries in Africa have national policies on traditional medicine, though implementation remains a problem [52]. The widespread use of herbal medicines for DSM in Africa suggests that governments need to give greater attention to the integration and regulation of the use of herbal medicines in the routine health system. In addition, rigorous research is also required to determine the potential effectiveness (or harm) of herbal medicine in diabetes management.

Another review finding was that most study participants, especially from Ghana, were engaged in 'healer shopping' ("the use of a second healer without referral from the first for a single episode of illness") [53]. Participants got involved in healer shopping for both biomedical and ethnomedical with the aim of finding affordable and effective treatments. The healer shopping practices in Ghana [34] and other parts of Africa is because participants misunderstood diabetes as a curable disease, suggesting a need for greater patient and family education in this area.

The review showed that, in the African context, most individuals with type 2 diabetes relied primarily on medication as a treatment strategy. Much less attention was given the other DSM activities such as exercise, foot care or diet. It appears that cultural beliefs, limited knowledge on diabetes management strategies and poverty were key barriers to the practice of DSM activities [54]. The review showed that there is a need to inform persons with diabetes about a range of affordable and practical forms of exercise to solve cited problems such as expensive gym fees. The practice of walking to the farm as indicated by rural dwellers could be encouraged, especially in settings where there is not available space for other forms of exercise.

The review showed that adherence to recommended dietary plans in order to control blood sugar levels was particularly a challenge because of the repetitive nature of the suggested meals and the perceived additional expense of the diabetic diet. Similar results in other studies have suggested that participants with limited varieties of foods in their local markets should be encouraged to eat food in moderation instead of avoiding them completely [39]. The review also showed that a degree of confusion existed among participants on the type and quantity of foods to eat [55]. It is therefore important that dietitians create more awareness and design consistent messages on food portioning and the effects of specific foods on blood sugar levels. Furthermore, these messages need to be adapted to local food preferences and habits and to the financial situation of patients

Consistent with reviews in other contexts [56], the evidence showed that social and family connections were both a facilitator and barrier to effective DSM. The family served as source of financial support in the purchase of medication and diabetic foods, but the family could also be a barrier to DSM due to the practice of communal cooking and eating. The extended family system where members cook and eat together is widely cherished in Africa and this makes it practically difficult for persons with diabetes to adhere to recommended dietary restrictions which may require the preparation of separate foods or eating at different times. This implies that persons with diabetes in Africa should not be managed in isolation of their social networks, but that significant others should be involved as an integral part of health promotion interventions. The strongly social and communal nature of eating in Africa suggests that health care professionals should try and engage key family members in health education and discussions around diet [57].

Another key finding of this review was that foot care appears to be a neglected practice in DSM in the African context. The review highlighted inappropriate foot care practices such as soaking the feet in hot water, applying hot water bottles and powder in-between toes, reflecting a lack of knowledge on foot care and unhelpful cultural beliefs (e.g. that foot ulcers were caused by witchcraft). Barriers to appropriate foot care were also associated with the regional context (e.g. participants in some studies indicated that the heat from the tropical weather prevented them from wearing closed shoes) [42]. Diabetic foot ulcers are complications of uncontrolled diabetes and, if not well managed, could lead to non-traumatic amputations and diminished quality of life of affected persons [58]. The dearth of literature on the foot care practices of persons with diabetes in Africa suggests a strong need for further studies in this area and for patient education to include culturally appropriate messages on diabetic foot care. Table 4

**Table 4. Recommendations to promote diabetes DSM related to barriers in the individual and community domain.**

| Barriers/Challenges | Recommendations for Policy, Practice or Future Research |
|---|---|
| Negative emotional states of persons living with diabetes | • Diabetes care initiatives should focus on the family and community to promote psychological resilience and wellbeing of persons living with diabetes.<br>• Healthcare professionals should provide emotional and psychological support to patients at different stages of their disease through effective dialogue |
| Patients combining herbal and biomedical medicines to treat diabetes | • More empirical work is needed to determine the potential effectiveness (or harm) of herbal medicine in diabetes management.<br>• Research work is also required to explore the potential for diabetes care algorithms to incorporate the use of clinically effective herbal preparations in order to give patients greater choice over potential treatment options |
| Poor financial situation of persons living with diabetes | • Clinicians should identify and introduce patients to less expensive local foods and other means of exercising other than expensive formal gyms. |
| Patients poor knowledge on diabetes as a chronic lifelong disease | • Culturally appropriate health education strategies for patient and family on diabetes should be done by health promotion officials. |
| Unhelpful cultural and superstitious beliefs | • Patient and family education required to demystify diabetes |
| Social and family connectivity/or ties | • Persons with diabetes should not be managed in isolation of their social networks, but that significant others should be involved as an integral part of health promotion intervention. This is particularly important to achieving dietary restrictions |

below presents a summary of the recommendations to promote DSM related to barriers in the individual and community domain.

## Experiences relating to the health system

The review showed that people with type 2 diabetes regarded health workers as a major source of support. Nonetheless, they complained that the health system and mode of care delivery was unfriendly and hard to access. In particular, participants seeking health care at hospitals complained of long waiting time at clinics and rushed consultations. A meta-synthesis in other contexts similarly identified that experiences of long queues and hurried consultations prevented persons with diabetes from discussing social or emotional challenges with service providers [56]. Our review showed that busy clinics, resulting in hurried consultations and insufficient time for persons with diabetes inhibited the potential for adequate patient education or correction of misconceptions and thus negatively impacted on DSM practices. This situation also does not allow ample time for persons with diabetes to share their emotional, social and cultural challenges with clinicians and this affects their ability to accept and be empowered to manage their condition [56]. Thus, this situation does not only affect the quality of care but also the self-management potential of persons living with diabetes. It has been asserted that inadequate time for person with diabetes and busy clinics is due to health system poor staffing issues [59] which could be resolved through deliberate policy directions. The impact of health system challenges on the promotion of DSM appears to be particularly significant in African countries where there is a deficit of over 2.4 million doctors [59]. Studies in Africa have indicated that delivering care to the growing numbers of persons with diabetes is problematic due to equipment shortages, excessive clinical workloads and lack of trained medical expertise [60]. To solve this problem, especially in rural Africa, rethinking health workforce and service delivery to take account of local resources, culture and socio-economic factors is important [61, 62]. Re-distribution of traditional healthcare roles is one potential solution [60]. For example, several studies conducted in African contexts have indicated that delivering diabetes care in a primary care setting, using diabetes trained nurses was effective in achieving glycaemic control [61, 63]. These nurse-led services were supported by patient education and diabetes treatment algorithms. They were shown to be accepted by the community and proved to be effective alternatives to overcome the shortage of trained physicians and other issues relating to caring for persons with chronic illnesses. Another potentially fruitful avenue for future research would be to explore group-based or online mechanisms of education and support.

Other health system factors highlighted in this review related to shortages of medications at clinics and of the high costs of medications. This made it difficult for persons with diabetes to maintain a constant refill of their medications and unable to follow the treatment regimen. The provision of additional government financial support (for example, in the form of insurance schemes and reliable medication supply) may be helpful in diabetes care. Studies in other contexts have similarly stressed that the poor financial status of persons with diabetes negatively affects DSM [64–66]. For example, in Africa, it has been argued that the monthly cost of treating diabetes can exceed the average salary [67]. Health ministries across Africa have acknowledged the impact and burden of chronic diseases, but very few nations currently have plans and policies on chronic diseases [68]. Therefore, it appears the individual with diabetes does not receive enough financial subsidy from governments. For instance, Ghana has a National Health Insurance policy but this does not include chronic conditions and diabetes medications are in the insurance policy's exemption list [69]. In order to enable effective DSM, there is an urgent need to develop insurance schemes that include chronic conditions like

**Table 5. Recommendations to promote diabetes DSM related to barriers within the health system.**

| Barriers/Challenges | Recommendations for Policy, Practice or Future Research |
| --- | --- |
| Busy clinics, long waiting time and hurried consultations | • Policy directives to restructure the health workforce and service delivery, taking into account local resources, culture and socio-economic factors is important. For instance, such policies could aim at re-distribution of traditional healthcare roles in order to increase the number of professionals in diabetes care.<br>• Further studies are also needed to explore the potentials of a group-based or online mechanism of education and support for persons living with diabetes |
| Shortage of medications at diabetes clinics | • Government's financial support and commitment to maintaining constant supply of medications at diabetes treatment centres is required |
| Costly diabetes medications | • Stakeholders in diabetes care, together with government should develop insurance schemes that include chronic conditions like diabetes. |

diabetes. Table 5 below presents a summary of DSM recommendations and barriers within the healthcare system.

## Strengths and weaknesses of the review

Our review has several strengths. Firstly, the review sought to specifically synthesize qualitative evidence on self-management experiences of persons with diabetes in Africa, rather than quantitative or experimental studies. The specific focus on Africa and upon qualitative evidence has illuminated issues that are particularly relevant to the African context and need to be considered in caring for persons with diabetes from this region. For example, family systems, cultural norms and practices related to food/diet, superstitious beliefs regarding the causes of diabetes and historical and ancestral use of herbal preparations were some of the issues that greatly influenced DSM practices of participants. Secondly, the review used transparent and rigorous methods, following the ENTREQ guidelines [19] and this allows for reproducibility of this study. A limitation of the review is that the included studies only represented seven countries in Africa, hence it is unclear to what extent the findings may be transferable to other African contexts. Given the scale and potential health impacts of diabetes in Africa, more research is needed to understand the experiences of persons with diabetes and to integrate these into appropriate health system responses.

## Conclusion

Our review demonstrates that the life style modifications required in DSM are problematic in the African context, primarily due to challenges that relate to the economic situation of persons with diabetes, unhelpful cultural beliefs and practices and unsupportive healthcare systems. The review shows a need to design culturally relevant health education strategies, to investigate the role of herbal medicines in DSM and to explore the role of non-medical practitioners in provision of care.

## Supporting information

**S1 File. Review potocol.**
(DOCX)

**S2 File. Search strategy.**
(DOCX)

**S3 File. List of countries covered by the WHO African region.**
(DOCX)

**S4 File. Quality appraisal of studies using JBI-QARI instrument.**
(DOCX)

**S5 File. Data extraction template.**
(DOCX)

**S1 Checklist. PRISMA 2009 check list.**
(DOCX)

## Acknowledgments

Thanks to Jonathan Bayuo and Mavis Mallory Mwinbam for helping to format and proof read this manuscript.

## Author Contributions

**Conceptualization:** Joseph Ngmenesegre Suglo.

**Data curation:** Joseph Ngmenesegre Suglo, Catrin Evans.

**Formal analysis:** Joseph Ngmenesegre Suglo, Catrin Evans.

**Investigation:** Joseph Ngmenesegre Suglo, Catrin Evans.

**Methodology:** Joseph Ngmenesegre Suglo, Catrin Evans.

**Project administration:** Joseph Ngmenesegre Suglo, Catrin Evans.

**Resources:** Joseph Ngmenesegre Suglo.

**Software:** Joseph Ngmenesegre Suglo.

**Supervision:** Joseph Ngmenesegre Suglo, Catrin Evans.

**Validation:** Joseph Ngmenesegre Suglo, Catrin Evans.

**Visualization:** Joseph Ngmenesegre Suglo, Catrin Evans.

**Writing – original draft:** Joseph Ngmenesegre Suglo.

**Writing – review & editing:** Joseph Ngmenesegre Suglo, Catrin Evans.

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
