## [Decision Letter · Decision Letter 0]

28 Jun 2020

PONE-D-20-14887

Factors influencing self-management in relation to type 2 diabetes in Africa: a qualitative evidence synthesis

PLOS ONE

Dear Dr. Suglo,

Thank you for submitting your manuscript to PLOS ONE. After careful consideration, we feel that it has merit but does not fully meet PLOS ONE’s publication criteria as it currently stands. Therefore, we invite you to submit a revised version of the manuscript that addresses the points raised during the review process.

There are several major issues in the manuscript that needs to be addressed.

We look forward to receiving your revised manuscript.

Kind regards,

Olayinka O Shiyanbola

Academic Editor

PLOS ONE

Journal Requirements:

2. Please ensure that you have addressed all items recommended in the PRISMA checklist including identifying the study as a meta-analysis or systematic review in the title.

Reviewers' comments:

Reviewer's Responses to Questions

**Comments to the Author**

1. Is the manuscript technically sound, and do the data support the conclusions?

Reviewer #1: Yes

2. Has the statistical analysis been performed appropriately and rigorously? 

Reviewer #1: N/A

3. Have the authors made all data underlying the findings in their manuscript fully available?

Reviewer #1: Yes

4. Is the manuscript presented in an intelligible fashion and written in standard English?

Reviewer #1: Yes

5. Review Comments to the Author

Reviewer #1: This paper describes a systematic review of qualitative studies of the experience of diabetes patients in African settings. The paper is appropriately structured and the rationale, methods and results are clearly described. The value of the published paper might be maximised if the key original and relevant findings from the evidence synthesis which directly or indirectly impact on self-management could be more clearly identified and if some of the specific implications of findings specific to the African context made more explicit.

Main issues

The Discussion could also be much clearer to what extent the finding are consistent with the evidence from the previous reviews and primary studies cited and which findings are either

1. Original and/or divergent and likely to be widely relevant to other settings

2. Original and/or divergent and likely that this is at least in part because they are specific to the African or country-specific culture or health care system.

For example high demand in clinics resulting in busy staff/queues/waiting times is presumably a wide issue for patients and not particularly specific as a barrier to DSM or something easy to address without fundamental change to diabetes care, whilst cultural beliefs about the aetiology and appropriate management of diabetes seems much more specific and relevant and might be addressed by adapting or expanding the existing diabetes education and support available in some regions and hospitals.

I am not sure whether the claim that the review focuses only on factors influencing self-management in type 2 diabetes is currently wholly justified but this is because the wider themes reported that may influence patient’s behaviour and diabetes care go beyond “self management” as defined within this paper as “life style modifications in areas such as exercise, diet, medication adherence, blood glucose monitoring and foot care”” eg busy clinics. The authors need to explain more explicitly how these factors may directly and indirectly influence patients’ capacity or motivation for these specific self care behaviours as they do for example in relating cultural obligations to diet and lack of appropriate facility to exercise practices, or costs to lack of self-monitoring or cultural practices and beliefs to drug compliance. Again this may produce some much more specific implication for potential changes to provision of education programmes or support for patients and families more generally.

The Discussion section also would be improved by the addition of

1. A summary of the strengths and weakness of the quality and scope of evidence identified

2. A summary of the strengths and weaknesses of this review.

This would potentially produce some more specific recommendation for further primary research and/or evidence synthesis to be included under “implications for further research”.

Minor issues:

Table 2 is not needed and could be removed or included as a appendix and replaced by a statement that six papers were found not to meet the inclusion criteria in terms of study population or study design.

The paper would benefit from careful proof reading throughout.

There are multiple misspellings in table 4 (“People with diabetess” “Using advise from diabetes class” "Shoulder and knee problems of personsons with diabetes”) and missing apostrophies (eg line 364 - people with diabetes’)

6. PLOS authors have the option to publish the peer review history of their article (what does this mean?). If published, this will include your full peer review and any attached files.

Reviewer #1: No

---

## [Author Response · Author response to Decision Letter 0]

22 Jul 2020

Response to Reviewer(s) PONE-D-20-14887

Factors Influencing Self-management in relation to Type 2 Diabetes in Africa: A Qualitative Systematic Review

PLOS ONE

Dear Editor,

Thank you for reviewing our manuscript and the points raised. We have revised the manuscript; taking into consideration PLOS ONE’s publication criteria and comments raised by the Academic Editor and Reviewer(s). Supporting files have been renamed to meet PLOS ONE style requirements. Also, all items in the PRISMA checklist have been addressed and the study has been identified as a systematic review in the title.

Below is a detail response to the various comments raised by the reviewer:

Reviewer’ comments: The value of the published paper might be maximised if the key original and relevant findings from the evidence synthesis which directly or indirectly impact on self-management could be more clearly identified and if some of the specific implications of findings specific to the African context made more explicit

Authors response: Thank you for this comment. We have tried to more clearly draw out the original and Africa-specific relevance of the review findings in various places in the review (as outlined in our more detailed responses below). For example, factors that serve as barriers and facilitators to self-management of diabetes are presented as a key part of table 3. Major findings affecting self-management such as misconceptions on the causes of diabetes, use of herbal medicines, cultural affinity for local foods, costly diabetes treatment and others are then also discussed again in more detail in the Discussion section, making more explicit the issues that are relevant to the African context.

Reviewers’ Comments: The Discussion could also be much clearer to what extent the finding are consistent with the evidence from the previous reviews and primary studies cited and which findings are either

1. Original and/or divergent and likely to be widely relevant to other settings

2. Original and/or divergent and likely that this is at least in part because they are specific to the African or country-specific culture or health care system

Authors’ Response: Thank you for this comment. We have tried to make these issues clearer by comparing our review findings with reviews from other contexts, while making clear the original and divergent findings specific to the African context. The revised discussion highlights that certain findings related to misconceptions on the causes of diabetes, costly diabetes treatment and the effects of family and social connectivity on self-management are comparable to previous review results - especially those from developing nations in other contexts. In the Discussion, we now highlight findings on family ties, health beliefs and cultural norms that are particularly relevant to the African context. The Discussion also points out widespread superstitious beliefs in Africa that diabetes is caused by punishment from ancestors and gods and how these beliefs influence their utilization of herbal medicines and faith healers. These findings are now discussed in relation to other systematic reviews and WHO interventions.

Reviewers’ Comments: For example high demand in clinics resulting in busy staff/queues/waiting times is presumably a wide issue for patients and not particularly specific as a barrier to DSM or something easy to address without fundamental change to diabetes care, whilst cultural beliefs about the aetiology and appropriate management of diabetes seems much more specific and relevant and might be addressed by adapting or expanding the existing diabetes education and support available in some regions and hospitals.

Authors’ Response: In the Discussion section of the revised paper, we have tried to bring out more clearly how busy clinics, long waiting time and queues, and the health system in general impacted on self-management. The discussion indicated that high demand on clinics negatively affected patient’s ability to discuss their emotional concerns with clinicians in order to receive the necessary support in the management of their diabetes. For instance, participants complained of not receiving education from clinicians and not being given ample time to share their challenges with healthcare givers. This situation may result in adoption of inappropriate self-management strategies due to misconstrued information from peers and other lay beliefs. Due to long queues and waiting time and non-availability of medications at clinics, some participants never visited the diabetes clinic again after their initial visit, also with the reason that healthcare workers did have enough time to listen to their concerns.

Reviewers’ Comments: I am not sure whether the claim that the review focuses only on factors influencing self-management in type 2 diabetes is currently wholly justified but this is because the wider themes reported that may influence patient’s behaviour and diabetes care go beyond “self management” as defined within this paper as “life style modifications in areas such as exercise, diet, medication adherence, blood glucose monitoring and foot care”” eg busy clinics. The authors need to explain more explicitly how these factors may directly and indirectly influence patients’ capacity or motivation for these specific self care behaviours as they do for example in relating cultural obligations to diet and lack of appropriate facility to exercise practices, or costs to lack of self-monitoring or cultural practices and beliefs to drug compliance. Again this may produce some much more specific implication for potential changes to provision of education programmes or support for patients and families more generally.

Authors Response: As rightly indicated by the reviewer, the review focused on factors influencing self-management (life style modification in areas of diet, exercise, medication adherence, blood glucose monitoring and foot care). In our revision, we have further documented more clearly in the results and discussion sections of this paper the behaviours and practices of patients in relation to these thematic self-care areas. We have also tried to explain more clearly the factors that influence these self-care practices as indicated in a detail response below:

In our revised discussion concerning diet, we added that some participants acknowledged that eating a healthy diet was necessary in controlling their diabetes. However, there was limited choices of foods in their local markets and this made the diabetic diet restrictive and repetitive to them. In addition, the costly nature of the diabetic diet, cultural value of local foods and others were also identified as influencing participants’ ability to follow a recommended dietary plan.

In the field of medication, we revised in our discussion that though some participants regarded diabetes treatment from hospital as the best option for treating their diabetes, the choice of treatment for many other participants depended on their perceived aetiology of the disease. Utilization of herbal medicines, faith healers and prayers as treatment options were mostly due to superstitious beliefs on the causes of diabetes. These beliefs made some participants to engage in multiple treatment modalities at the same time.

Exercise was yet another area of self-care we revised. We added that participants complaining of expensive gyms and lack of space to exercise should be introduced to diverse ways of exercising. As an example, rural farmers could be encouraged to walk to farms as they themselves suggested.

Concerning blood sugar monitoring, we further made it more clear under the results section that most participants were not able to keep a diary of their blood glucose because they did not have personal glucometers at home. This was due to the expensive nature of the machine and some participants therefore had to rely on their bodily signs to guess their blood sugar level.

Both in the results and discussion sections, we have highlighted the foot care practices of respondents. The discussion explored further on the inappropriate foot care practices of participants which was partly due to their insufficient knowledge on foot care and misconceptions. The heat from the tropical weather was also noted as a contributory factor for non-wearing of close shoes. Some participants with foot problems have never visited the clinic and have never received any health education on foot care

Reviewers’ Comments: The Discussion section also would be improved by the addition of

1. A summary of the strengths and weakness of the quality and scope of evidence identified

2. A summary of the strengths and weaknesses of this review.

Authors Response: Thank you for this. The strengths and weaknesses of our review and how this might have affected the findings have been summarized and presented as part of the revised paper.

Reviewers’ Comments: Table 2 is not needed and could be removed or included as an appendix and replaced by a statement that six papers were found not to meet the inclusion criteria in terms of study population or study design.

Authors Response: List of excluded studies (Table 2) has been deleted from the revised paper as recommended by the reviewers

Reviewers’ Comments: The paper would benefit from careful proof reading throughout. There are multiple misspellings in table 4 (“People with diabetess” “Using advise from diabetes class” "Shoulder and knee problems of personsons with diabetes”) and missing apostrophies (eg line 364 - people with diabetes’)

Authors Response: Proof reading of the manuscript has been done and mistakes corrected

---

## [Decision Letter · Decision Letter 1]

2 Sep 2020

PONE-D-20-14887R1

Factors Influencing Self-management in relation to Type 2 Diabetes in Africa: A Qualitative Systematic Review

PLOS ONE

Dear Dr. Suglo,

Thank you for submitting your manuscript to PLOS ONE. After careful consideration, we feel that it has merit but does not fully meet PLOS ONE’s publication criteria as it currently stands. Therefore, we invite you to submit a revised version of the manuscript that addresses the points raised during the review process.

ACADEMIC EDITOR:

Further discussion on future directions, clinical implications, and practical recommendations for improving self-management in Africa.

We look forward to receiving your revised manuscript.

Kind regards,

Olayinka O Shiyanbola

Academic Editor

PLOS ONE

Additional Editor Comments (if provided):

The authors should further specify and discuss future recommendations and directions for improving self-management of Type 2 diabetes in African countries.

This information could be based on what they found in their qualitative systematic review.

Reviewers' comments:

Reviewer's Responses to Questions

**Comments to the Author**

1. If the authors have adequately addressed your comments raised in a previous round of review and you feel that this manuscript is now acceptable for publication, you may indicate that here to bypass the “Comments to the Author” section, enter your conflict of interest statement in the “Confidential to Editor” section, and submit your "Accept" recommendation.

Reviewer #1: All comments have been addressed

2. Is the manuscript technically sound, and do the data support the conclusions?

Reviewer #1: Yes

3. Has the statistical analysis been performed appropriately and rigorously? 

Reviewer #1: N/A

4. Have the authors made all data underlying the findings in their manuscript fully available?

Reviewer #1: Yes

5. Is the manuscript presented in an intelligible fashion and written in standard English?

Reviewer #1: Yes

6. Review Comments to the Author

Reviewer #1: (No Response)

7. PLOS authors have the option to publish the peer review history of their article (what does this mean?). If published, this will include your full peer review and any attached files.

Reviewer #1: No

---

## [Author Response · Author response to Decision Letter 1]

3 Oct 2020

Response to Reviewer(s) PONE-D-20-14887R1

Factors Influencing Self-management in relation to Type 2 Diabetes in Africa: A Qualitative Systematic Review

PLOS ONE

Dear Editor,

Thank you for reviewing our manuscript and the points raised. We have revised the manuscript; taking into consideration PLOS ONE’s publication criteria and comments raised by the Academic Editor.

Detail response to the reviewers’ comments is indicated below:

Editor’s Comments: Further discussion on future directions, clinical implications and practical recommendations for improving self-management in Africa

Authors Response: Thank you for these points raised. Within the discussion section of the revised paper, we have added in substantive further points regarding the clinical implications of our findings, whilst also giving additional recommendations for interventions that could be developed and evaluated to improve self-management.

To make our recommendations for policy and practice clearer, we have also summarised these in tables 4 and 5 in the manuscript.

---

## [Editor Report · Decision Letter 2]

6 Oct 2020

Factors Influencing Self-management in relation to Type 2 Diabetes in Africa: A Qualitative Systematic Review

PONE-D-20-14887R2

Dear Dr. Suglo,

We’re pleased to inform you that your manuscript has been judged scientifically suitable for publication and will be formally accepted for publication once it meets all outstanding technical requirements.

Kind regards,

Olayinka O Shiyanbola

Academic Editor

PLOS ONE
---

## [Editor Report · Acceptance letter]

8 Oct 2020

PONE-D-20-14887R2 

Factors Influencing Self-management in relation to Type 2 Diabetes in Africa: A Qualitative Systematic Review 

Dear Dr. Suglo:

I'm pleased to inform you that your manuscript has been deemed suitable for publication in PLOS ONE. Congratulations! Your manuscript is now with our production department. 

Kind regards, 

on behalf of

Olayinka O Shiyanbola 

Academic Editor

PLOS ONE